# Up to a 15-Year Survival Rate and Marginal Bone Resorption of 1780 Implants with or without Microthreads: A Multi Center Retrospective Study

**DOI:** 10.3390/jcm12062425

**Published:** 2023-03-21

**Authors:** Ji-Hwan Oh, Se-Wook Pyo, Jae-Seung Chang, Sunjai Kim

**Affiliations:** 1Graduate School, College of Dentistry, Yonsei University, Seoul 03722, Republic of Korea; 2Department of Prosthodontics, Gangnam Severance Dental Hospital, College of Dentistry, Yonsei University, Seoul 06273, Republic of Korea

**Keywords:** marginal bone resorption, microthreads, multicenter, platform-switching, survival rate

## Abstract

The effect of microthreads at the implant neck on the amount of marginal bone resorption is controversial. This multicenter retrospective study compared the implant survival rate and amount of marginal bone resorption between two platform-switching internal connection implant systems with or without microthreads. Patient-related (age and sex), surgery-related (implant installation site, type, diameter, and length), and prosthesis-related (prosthesis type) data were collected from patient charts from the implant placement surgery to the final recall visit. A total of 1780 implants, including 1379 with microthreads and 401 without microthreads, were placed in 804 patients. For implants with and without microthreads, the longest follow-up period after prosthesis delivery was 15 and 6 years, respectively. Twenty implants failed during the 15-year follow-up period (98.8% survival rate) due to failed osseointegration, peri-implantitis, implant fractures, and non-functioning implants. The mean marginal bone loss was < 0.1 mm for both implant systems at the 1-year follow-up and 0.18 mm and 0.09 mm at the 4-year follow-up for implants with and without microthreads, respectively, without statistical significance. Microthreads did not significantly affect the amount of marginal bone loss or the implant survival rate for implants with an internal connection with a platform-switching design.

## 1. Introduction

Implant-supported fixed restoration is a predictable treatment modality for a partially edentulous dentition, with a 5-year implant survival rate of >97% and a 15-year survival rate of >90% [1,2,3,4]. The amount of marginal bone loss (MBL) is frequently used to evaluate the results of implant treatment [5,6,7]. Furthermore, it has been reported that the first-year MBL after implant placement is an important factor in the success or failure of implant treatment [8]. The amount of MBL is influenced by patient-related factors such as the patient’s history of periodontitis, smoking status, and plaque control ability, as well as implant-related factors such as implant connection and geometric configuration [9,10,11,12,13].

Among the implant-related factors, microthreads and platform switching were meticulously studied by many investigators. Platform switching is an efficient method for reducing the amount of MBL [14]. A meta-analysis study showed that platform switching did not affect the implant survival rate but reduced MBL significantly as compared with platform-matched implant–abutment connections [5]. A recent systematic review analyzed the MBL of 426 platform-switched and 411 platform-matched implant prostheses and concluded that platform switching resulted in a significantly smaller MBL. In that review, the greater the mismatch between the implant and abutment diameter, the lower the MBL [15]. With the introduction of rough surfaces and the platform-switching concept, the amount of MBL was significantly reduced as compared with that of the earlier implants [6].

Compared to platform switching, previous studies reported controversial results with microthreads design. Finite element analysis (FEA) studies have shown that the microthread design of the implant neck effectively reduced the stress on the marginal bone [9]. Bratu et al. used an implant with microthreads and a rough surface in the implant neck area or a machined surface without microthreads to restore missing mandibular posterior teeth. They reported significantly less MBL when the implants had microthreads and rough surfaces [10]. A 3-year prospective study also reported that implants with microthreads had significantly less MBL compared to the implants without microthreads [11]. Contrary to the studies above, several studies have questioned the need for microthreads. An FEA study demonstrated increased crestal stress around microthreaded implants but decreased crestal stress with platform-switched implants without microthreads [16]. Shin et al. compared the amount of MBL using three different implants: those with microthreads and a rough surface, those without microthreads but with a rough surface, and those without microthreads and with a polished surface [17]. Their 1-year study concluded that implants with microthreads and a rough surface had significantly less MBL than implants with a polished surface. However, when the implant had a rough surface, there was no significant difference in MBL according to the presence or absence of microthreads. Van de Velde also reported no significant difference in MBL between rough-surface implants with microthreads and rough-surface implants without microthreads [18]. These studies indicate that platform-switching with a rough surface sufficiently prevented MBL even without the aid of microthreads.

Friction-fit internal connection implants are popular in clinical implant dentistry, and different internal taper angles are used for different implant systems. It is reported that friction-fit internal connection implants resulted in a higher incidence of implant fracture rate compared to external connection implants [19]. Even an implant fracture is not a common event; however, the fractured implant should be removed and regarded as a failure when it does occur. A recent study reported that the incidence of an implant wall fracture was influenced by the degree of internal taper because the smaller the taper, the thinner the implant wall thickness [20,21].

The current study evaluated the implant survival rate, causes of implant failures, and amount of marginal bone resorption for up to 15 years, using two internal connection implant systems, with or without microthreads. Both implant systems had the platform-switching design and the same surface treatments from the same manufacturer. The null hypothesis tested was that there was no significant difference between the implants with or without microthreads in the amount of marginal bone loss as well as survival rate.

## 2. Materials and Methods

This retrospective study was approved by Gangnam Severance Hospital and Yonsei University (Institutional Review Board approval number 3-2020-0181). This study included all partially edentulous patients who received implant-supported fixed dental prostheses at the Department of Prosthodontics, Gangnam Severance Dental Hospital, and Hayan Dental Clinic between 2003 and 2020. One clinician at each institute performed the surgeries and restorative procedures. The exclusion criteria were as follows: (1) psychological disorder, (2) uncontrolled diabetes mellitus, (3) immune suppression, (4) previous radiotherapy to the head and neck region, or (5) parafunctional oral habits, such as teeth clenching and bruxism, (6) implant systems other than the current study design, and (7) immediately loaded implants. Inclusion criteria were that: (1) the patient’s age was greater than 18 years old; (2) all patients met the diagnostic criteria for a dentition defect; (3) the patients had no contraindications to surgery; (4) informed consent was provided; and (5) female participants were non-pregnant, non-lactating, and not menstruating.

The study used two different implant systems from the same manufacturer. Both implants had the same surface topographies. One had a platform-switching design with microthreads at the implant neck (IT; Warantec, Seoul, South Korea). The implant–abutment connection had a 7° angle to the long axis. The other had a platform-switching design without microthreads and an 11° internal taper angle (IU, Warantec) (Figure 1).

Handwritten and electronic charts from the implant placement surgery to the final periodic recall visits were reviewed to collect patient-related (age and gender), surgery-related (implant installation site, implant type, implant diameter, and implant length), and prosthesis-related (prosthesis type) information.

Implants were categorized according to their diameter (<4 mm, 4–5 mm, and >5 mm), length (<8.5 mm, 8.5–10 mm, and >10 mm), and design (IT or IU). The installation site was divided into the maxilla, or mandible, and anterior or posterior dentition. The prosthesis type was classified into three categories based on the adjacent dentition or restorations for comparing the amount of MBL, as follows: single, a single implant-supported restoration between teeth or the most distal crowns; consecutive, whether splinted or not, an implant-supported crown next to an implant crown or crown; and fixed partial dentures (FPDs), implant-supported dentures with pontics, without adjacent implant crowns. Implant survival was defined as follows: the implant remained in the patient’s mouth, and the restoration functioned normally during the last periodic visit. Therefore, removed and buried (submerged and not functional) implants were classified as failures.

IU implants have only recently been introduced to clinical use as compared with IT implants; therefore, IU implants have had shorter follow-up recalls than IT implants. Cumulative survival rates were calculated separately for each implant system. Periapical radiographs, which were obtained at the delivery of the final prostheses and at each periodic recall visit, were used as baselines to analyze the amount of peri-implant marginal bone changes at each recall visit. When periapical radiographs were not taken every year, a 6-month interval was used to input only a single measured value for a single time interval. A customized device was not used for each of the periapical radiographs. Instead, a commercial extension cone parallel instrument (XCP; Dentsply, York, PA, USA) was used for each periapical radiograph. Image processing and analysis software (Image J 1.53, https://imagej.nih.gov/ij/download.html, accessed on 2 March 2023) was used for all measurements. A distance calibration was performed for every radiograph to compensate for angular distortion. The amount of MBL was defined as the distance between the outer edge of the implant platform and the most coronal bone-to-implant contact point on both the mesial and distal sides of the implant. All measurements were performed by a single operator. Clinically, the diagnosis of peri-implantitis requires the following: (1) bleeding on probing or suppuration; (2) more than 6 mm of probing depth; and (3) more than 3 mm of marginal bone loss compared to the initial bone level. In the current study, previous probing depth was not always obtained; therefore, an implant with ≥3 mm of marginal bone loss and bleeding on probing was used to diagnose peri-implantitis. A lifetime table was used to summarize the cumulative survival rate for each period for the IT and IU implants. Due to the differences in periods between the two implants, a simple arithmetic comparison was performed at each follow-up period. SAS version 9.4 (SAS Institute, Cary, NC, USA) was used for statistical analysis. A linear mixed model was used to compare the effects of variables on the amount of marginal bone. The Mann–Whitney U test was used to compare the amount of MBL at each time period between the implant systems. A significance level of 95% was used for all statistical analyses.

## 3. Results

### 3.1. Description of the Implant and Patient Cohort

A total of 1780 implants, including 1379 IT and 401 IU implants, were placed in 804 patients, with an average of 2.2 implants per patient. The mean age of patients was 58.5 years (standard deviation ± 13.7). The longest follow-up period was 15 years after delivery of the prostheses for IT implants and 6 years for IU implants. Table 1 presents the demographic data of the patients and implants.

### 3.2. Survival Rate

Survival was considered when the implant remained and functioned in the oral cavity at the time of the final examination. Even if <1 year had elapsed after prosthesis placement, if there was an apical radiograph taken at the 6-month examination, the data were included in the survival rate analysis but not in the assessment of marginal bone changes.

For IT implants, 16 failed during the follow-up up to 15 years after surgery, and the implant survival rate was 98.8%. Four IU implants were removed within 6 years of follow-up after delivery of the prostheses, and the implant survival rate was 99.0%. Table 2 presents the detailed data on the failed implants. Among the five early failures, two implants were associated with the infection of the graft material, which was deproteinized bovine bone mineral.

### 3.3. Bone Level Comparison

Marginal bone changes were evaluated for 1309 IT and 325 IU implants. Many cases did not meet the exact 1-year interval recall; therefore, the recall periods were divided into 1, 2, 4, and 6+ years (Table 3).

The effect of each variable on the demographic data was evaluated; however, no single factor was found to have a significant effect on MBL. The results of linear mixed model is presented as Appendix A. 

### 3.4. Incidence of Peri-Implantitis

Because the plaque index and probing depth were not recorded for all implants, peri-implantitis was diagnosed using a case of more than 3 mm of marginal bone loss compared to the initial bone level, combined with bleeding on probing. Based on the above, the incidence of peri-implantitis was 2.0% at 4 years and 2.8% at 6+ years of recall.

## 4. Discussion

This study evaluated the clinical results of two implant systems with different macrogeometries for up to 15 years. The two implant systems had the same surface roughness and platform-switching design; however, one implant had microthreads, while the other did not. Another significant difference between the implant types was the degree of the internal connection angle between the implants and abutment. The implant with microthreads had a 7° angle, while the implant without microthreads had an 11° angle to the long axis of the implant. During the 15 years, 20 implants were removed, resulting in a 98.9% survival rate for implants with microthreads and a 98.8% survival rate for implants without microthreads. These survival rates were comparable to those reported in previous studies [1,2,3,4]. There were four main reasons for implant failure: failure of osseointegration, implant fractures, non-functioning implants, and implants with peri-implantitis. The amount of MBL was not significantly different between the two types of implants; thus, the null hypothesis was rejected.

The most common reason for implant removal is osseointegration failure. Ten implants in our study were removed for this reason and classified as early failures. In addition to these 10 early failures, 10 implants were removed or were non-functional after prosthesis delivery.

Five implant fractures occurred in four patients, resulting in a 0.3% implant fracture rate. Three fractured implants were used to restore splinted crowns, or FPDs, and the other two were single-implant restorations (Figure 2).

Interestingly, one patient had two implant fractures, one in each of the left and right mandibular posterior quadrants. The earliest fracture in our patients occurred after 6 years of function, followed by two fractures that occurred after 7 years, one after 8 years, and one after 9 years. All fractures occurred in the posterior teeth, which produced greater occlusal force compared to the anterior teeth. At the time of implant fracture, three implants had >3 mm of marginal bone resorption, and the remaining two implants had <1.5 mm of bone loss. Manzoor et al. reported the stability of implant–abutment assemblies for simulated MBLs of 0, 1.5, 3, and 4.5 mm and reported frequent horizontal implant fractures for a simulated MBL of >3 mm [22]. They concluded that the cross-sectional geometry of the implant changed from a solid cylinder to a hollow tube by approximately 3 mm, resulting in a mechanically vulnerable area. Therefore, maintaining the marginal bone level is crucial for the biomechanical sustainability of the implant–abutment assembly and for long-term implant function.

A 9-year study reported a relationship between MBL and implant fracture. When marginal bone resorption was <50% of the implant length, vertical or horizontal implant fractures occurred mainly at the implant–abutment junction. However, when bone resorption occurred in >50% of the implant length, the incidence of a horizontal implant fracture was remarkably high in the area far beyond the implant–abutment junction [23]. No horizontal implant fracture was found in the current study; instead, all fractures were limited to the implant–abutment connection area. All the fractured implants had a diameter of 4.3 mm and a 7° internal connection angle, i.e., they were microthreaded implants. No fracture was found in microthreaded implants with greater diameters or in implants without microthreads.

A recent study reported a 3.5% implant fracture rate, which was considered quite high [24]. The authors attributed the high incidence of fracture to small-diameter implants (3.6 mm in diameter). Small-diameter implants have a thinner outer wall, which is a risk factor for implant fracture.

In the current study, the incidence of implant fracture was 0.3%, similar to that reported in other previous studies [23,25]. The current study used implants with two different internal connection angles (7° and 11°). Jin et al. observed 12,538 implants with two different internal connection angles over an 8-year period. There was a significant difference in implant fracture incidence between the 7.5° and 5.7° connection angles [21]. The smaller internal connection angle resulted in a significantly higher fracture incidence. The thinnest area of the 4.3-mm-diameter microthreaded implant was 0.39 mm, while the thinnest outer wall thickness of the 4.0- and 4.5-mm-diameter implants without microthreads was 0.33 mm and 0.54 mm, respectively. Although the thinnest wall thickness of the 4.0-mm-diameter non-microthreaded implant was 0.33 mm, which was thinner than the 0.39 mm of the 4.3-mm-diameter microthreaded implant, 87% of the non-microthreaded implants used in the posterior region had a diameter of ≥4.5 mm, and 4.0-mm-diameter non-microthreaded implants were used only for the premolars. In contrast, 82% of the microthreaded implants in the posterior quadrants had a diameter of 4.3 mm.

There was one case in which the implant did not have peri-implantitis or an implant fracture but was not functional. A distal cantilever FPD using implants in the second premolar and first molar areas functioned for 5 years without any complications. The prosthesis fell out with a fracture of both abutment screws after 5 years of function. The screw fragment in the second premolar implant was removed; however, it was impossible to remove the screw fragment from the first molar implant. The first molar implant was submerged, and a short implant was placed in the second molar area to fabricate a 3-unit FPD. The submerged implant was classified as a nonfunctioning implant and was included in the failed implant category.

Previous studies have concluded that distal cantilever prostheses did not increase marginal bone resorption [26,27]; however, cantilever FPDs resulted in a greater risk of prosthetic complications [28,29]. A meta-analysis reported the 5-year estimated incidences of prosthetic complications. The incidence of screw loosening was the highest at 8.2%, followed by abutment/screw fracture at 2.1% and implant fracture at 1.3% [30]. A 95-month retrospective study reported the implant fracture rate based on 10,099 implants. Remarkably, the fracture risk increased by 247.6% in cantilevered prostheses [31]. In the current study, cantilever FPDs did not introduce implant fractures, although if abutment screw fracture fragments could not be retrieved from the implant, it was the same as in the case of implant failure. When use of a distal cantilever FPD is inevitable, it is recommended that a reduced occlusal table size be used to decrease the actual occlusal force applied to implants, abutments, and abutment screws.

Two implant types used in the current study resulted in comparable amounts of MBL with previous studies using different microthreaded implants [10,11]. Since six years was the most extended observation period for implants without microthreads, a longer observation period might lead to different results. However, considering that bone resorption around implants usually occurs in the early stages, it can be concluded that even without microthreads, platform-switching could efficiently minimize marginal bone loss.

Four implants were removed because of peri-implantitis. A single mandibular molar implant was the earliest implant removed (after 3 years of function). This implant resulted in an MBL of >2 mm at the first-year follow-up and of >3.5 mm, with bleeding on probing, in the third year. Even with efforts to treat peri-implantitis, the patient was not satisfied with the implant treatment; therefore, the implant was removed and replaced with a new implant. The new implant showed no biological or technical complications. Three more implants were removed due to peri-implantitis after 9, 11, and 14 years of function. Peri-implantitis is a plaque-associated disease that is closely associated with poor plaque control and poor maintenance care [32]. Plaque control is influenced by the patient’s ability to clean the oral environment but is also significantly influenced by the contour of the implant prosthesis. Ravida et al. compared the amount of MBL for three different prosthetic modalities to replace three consecutively missing posterior teeth. The modalities were forty of non-splinted single crowns (NSC), 52 of 3-unit splinted crowns (SC), and 53 of 3-unit FPDs over two implants (FPD). The implant survival rates were 100% for the 3-unit FPDs, 92.5% for the NSCs, and 88.5% for the SCs, and the frequency of peri-implantitis was 16.7% for SCs, 7.8% for NSCs, and 2.8% for FPDs [33].

It is assumed that NSCs and FPDs are favored over SCs for interproximal cleansing. In the current study, the three implants removed due to peri-implantitis at 9, 11, and 14 years had SCs and used three or more consecutive implants. The ease of cleaning the interproximal area may be associated with implant removal. It is wise to have an adequate distance between implants for easy proximal cleansing when restoring consecutively missing teeth in the posterior dentition. The inter-implant distance also influences the shape of the transmucosal abutment. The increased inter-implant distance results in a greater emergence angle of the transmucosal abutment, which is associated with the risk of peri-implantitis [34]. Therefore, it is essential to determine the implant position based on the final prosthesis design to provide easy cleansing access.

The current study had some limitations. As mentioned above, a direct comparison was not possible due to the different service periods of the two implant systems. Therefore, the comparison of bone resorption between the two implants was evaluated only for 6 years, which was the most extended follow-up period for non-microthreaded implants. This study was conducted by describing the clinical results of the two implant systems rather than making a direct comparison between them. The incidence of peri-implantitis in the current study was low compared to the previous studies. It is assumed that these low incidences were due to insufficient information about peri-implant soft tissue conditions. The current study was more focused on the survival rate and the amount of MBL than on peri-implant soft tissue health. Despite the abovementioned limitations, this study was significant in that it evaluated the mid-to-long-term results of two different implant systems rather than performing a short-term evaluation. Future studies should include direct comparisons between implants as well as prosthetic complications for each implant system.

## Figures and Tables

**Figure 1 jcm-12-02425-f001:**
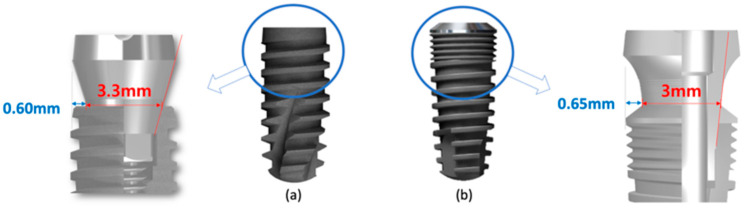
Two implant types were used in the current study. (**a**) implant without microthread (IU); (**b**) implant with microthreads (IT).

**Figure 2 jcm-12-02425-f002:**
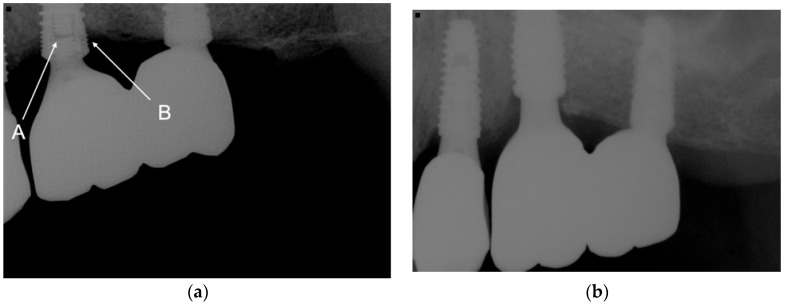
(**a**) The fractured abutment screw and fractured implant neck inside the implant led to the loss of the implant at the maxillary left first molar. A indicated fractured abutment screw. B indicated fractured implant wall. (**b**) The fractured implant was removed, a new implant with a 5-mm diameter was placed, and a new splinted crown was delivered.

**Table 1 jcm-12-02425-t001:** Demographic data of patients and the included implants.

Variables		N
Implant types	IT	1379
	IU	401
Sex	Male	799
	Female	981
Diameter	<4 mm	98
	4–5 mm	1444
	>5 mm	239
Length	<8.5 mm	63
	8.5~10 mm	1591
	>10 mm	126
Prosthesis type	Single	472
	Consecutive	1032
	Bridge	276
Maxilla	Anterior	193
	Posterior	738
Mandible	Anterior	38
	Posterior	793

**Table 2 jcm-12-02425-t002:** Detailed data on failed implants.

Time of Failure	Patient	Age	Sex	Position	Implant Type	Diameter	Length	Prosthesis Type	Cause of Failure
BDP	1	75	M	I27	IT	4.3	10	-	Graft infection
BDP	2	59	F	I36	IT	4.3	10	-	Graft infection
BDP	3	71	M	I37	IU	5	8.5	-	Failure of osseointegration
BDP	4	55	F	I22	IT	4.3	11.5	-	Failure of osseointegration
BDP	5	56	M	I14	IT	4.3	13	-	Failure of osseointegration
<1 year	1	75	M	I26	IT	4.3	11.5	Consecutive	Failure of osseointegration
<1 year	6	54	F	I27	IT	4.3	10	Consecutive	Failure of osseointegration
<1 year	7	72	F	I46	IT	4.3	8.5	Single (most distal)	Failure of osseointegration
1 year	8	50	F	I47	IU	4.5	10	Single (most distal)	Failure of osseointegration
2 years	9	66	M	I26	IU	4.5	10	Consecutive (most distal)	Failure of osseointegration
3 years	10			37	IT	4.3	10	Single (most distal)	Peri-implantitis
5 years	11	62	M	I26	IT	4.3	10	Consecutive	Abutment neck fracture and implant fracture
5 years	12	66	M	I36	IT	4.3	10	Consecutive (cantilever)	Remained fractured fragment, implant buried
8 years	13	59	M	I46	IT	4.3	10	Single (most distal)	Abutment screw fracture and implant fracture
7 years	14	69	F	I36	IT	4.3	10	Consecutive	Implant fracture and peri-implantitis
7 years	15	29	F	I47	IT	4.3	8.5	Single (most distal)	Abutment neck fracture and implant fracture
9 years	16	60	F	16	IT	4.3	11.5	Consecutive	Peri-implantitis
9 years	14	71	F	I45	IT	4.3	10	FPD (mesial implant)	Implant fracture and peri-implantitis
11 years	17	60	F	I16	IT	4.3	11.5	Consecutive	Peri-implantitis
14 years	18	52	F	I14	IT	4.3	10	Consecutive	Peri-implantitis

**Table 3 jcm-12-02425-t003:** Means and standard deviations of each follow-up period.

	Mean (Standard Deviation) of Mesial Marginal Bone Loss				Mean (Standard Deviation) of Distal Marginal Bone Loss			
	1 Year	2 Years	4 Years	6+ Years	1 Year	2 Years	4 Years	6+ Years
IT	0.06 (0.2)	0.10 (0.27)	0.18 (0.45)	0.29 (0.6)	0.08 (0.24)	0.12 (0.30)	0.18 (0.45)	0.23 (0.55)
IU	0.04 (0.12)	−0.07 (0.19)	0.09 (0.20)	0 (0.0)	0.05 (0.15)	0.07 (0.20)	0.08 (0.18)	0 (0.0)
	*p* > 0.05							

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
