# Peer review of "Up to a 15-Year Survival Rate and Marginal Bone Resorption of 1780 Implants with or without Microthreads: A Multi Center Retrospective Study"

_jcm, 2023, doi:10.3390/jcm12062425_

Round 1

Reviewer 1 Report

The manuscript entitled as “Marginal Bone Resorption and up to 15-Year Survival Rate of 1,780 Implants with or without Microthreads: A Multicenter Retrospective Study” tackles differences about implant survival, reasons of failure and MBL between two implant systems, with and w/o microthreads. The implant systems have the same surface topography, however the internal connection itself has some differences, which does not compromise the comparison. Bellow, the authors can find some suggestions and concerns.

Abstract:

Very adequate, objective and contains all essential information.

Introduction:

Contains relevant information of the concepts (microthread; switch platform and internal connections) and outcomes from other studies about these topics. It is very adequate as well. However, in some parts of the text, the writing is more discussion-like than introduction, that is, too detailed about specific study from the literature. I suggest to the authors summarizing the description of some studies cited, focusing on the conclusion itself than the study methods.

Another point, the authors competently described the concepts, what is already known from the literature and the objectives of the present study. However, in the reviewer’s opinion, it is still lacking a phrase describing the need of the present study, that is, the importance of a retrospective study focused on the microthread.

Methods:

The authors described the implants and platforms (with microthread, with an octagon internal connection and w/o microthread, with a hexagon internal connection), exclusion criteria, both centers and period. Periapical radiographies with distance calibration were used and a single operator measured all MBL. Authors pointed out the reason for the significant discrepancy in implant numbers between systems.

In the introduction, the authors describe that smoking habit influences in MBL. However, no information was given in the methods regarding it, that is, if smokers were included and, if they were, how the authors dealt with this data. Being such an important factor, the authors should disclaim it. Another disclaim that might be relevant is the surface treatment – Do both systems have the same surface treatment?

Results:

Implant loss occurred predominantly in patients with more than 60 years old. I suggest the inclusion of the mean age of the 804 patients in demographics.

Fixed partial dentures (FPDs) are sometimes abbreviated as FDPs throughout the text.

Table 3: the values present are negative. In a first sight, the reviewer thought that the negative value meant that the loss was negative, which means, a bone gain. After some thought, I realized that the value meant the actual loss. I suggest a fine tuning in the table to avoid misinterpretation (e.g., add information in the table description).

The survival rate obtained in the present study is superior to the average survival rate described in the literature. Seeing the exclusion criteria (especially bruxism), this finding makes sense.

No subgroups inside IT and IU, regarding MBL, were shown. The authors claimed that no single factor was found to have a significant effect on MBL. Nevertheless, it would be useful (as supplementary material) to have the values of subgroup division.

Figure 2: Add A and B to the figures to follow the legend. In addition, I suggest pointing out (with an arrow) the fracture or adding a zoomed picture to facilitate the appreciation of the fracture.

Discussion:

Authors discussed the causes of implant failure, lack of osseointegration, fractures, peri-implantitis and non-functional implants. The discussion is well-written and tackled the reason and literature comparison. However, In the reviewer’s view, the discussion is significantly more focused on implant failures than MBL. Since the objective of the present study embraces both outcomes, I suggest the addition of a paragraph about MBL.

Implant fractures. I suggest disclaiming if patients have some non-diagnosed parafunctional habit (e.g., just clenching, without grinding) and, after the first fracture, if some action was taken (e.g., bite guard or Botox application).

Author Response

Dear reviewer. First of all, thanks for the review. I really appreciated. Some rephrased part may not meet your standard due to the nature of retrospective study. Soft tissue information was not fully recorded. The recall extensions were different between the IU and IT implant. However, we tried to follow your comments and advices as possible.

Introduction

Contains relevant information of the concepts (microthread; switch platform and internal connections) and outcomes from other studies about these topics. It is very adequate as well. However, in some parts of the text, the writing is more discussion-like than introduction, that is, too detailed about specific study from the literature. I suggest to the authors summarizing the description of some studies cited, focusing on the conclusion itself than the study methods.

Thanks for the comments. I removed superfluous sentences to simplify the introduction part. For example, I deleted the concept of platform switching and detailed explanations about microthreads. Since I removed those sentences, I think the introduction part is more compact and easier to read. Thanks again for your good advice.

Another point, the authors competently described the concepts, what is already known from the literature and the objectives of the present study. However, in the reviewer’s opinion, it is still lacking a phrase describing the need of the present study, that is, the importance of a retrospective study focused on the microthread.

Thanks for your valuable comments. I re-arranged the effect of platform-switching and microthreads to introduce the controversy about microthreads. Therefore, I emphasized the aim and null hypothesis of the current study.

Method

The authors described the implants and platforms (with microthread, with an octagon internal connection and w/o microthread, with a hexagon internal connection), exclusion criteria, both centers and period. Periapical radiographies with distance calibration were used and a single operator measured all MBL. Authors pointed out the reason for the significant discrepancy in implant numbers between systems.

In the introduction, the authors describe that smoking habit influences in MBL. However, no information was given in the methods regarding it, that is, if smokers were included and, if they were, how the authors dealt with this data. Being such an important factor, the authors should disclaim it.

Thanks for the great comments. You are right. The smoking habit is a critical factor on peri-implantitis.Unfortunately,due to the multi-center and retrospective nature, very important patient information such as the smoking habit was not recored for all the patients included  therefore, we could not address the effect of smoking habit in the study. We are trying to record all important patient data in the future studies. Thanks.

Another disclaim that might be relevant is the surface treatment – Do both systems have the same surface treatment?

Yes. Both implants have the same surface topography, and I mentioned in the materials and methods section. Thanks

Results

Implant loss occurred predominantly in patients with more than 60 years old. I suggest the inclusion of the mean age of the 804 patients in demographics.

Thanks for the comments. We added the mean age and standard deviation of patients.

Fixed partial dentures (FPDs) are sometimes abbreviated as FDPs throughout the text.

Thanks for the comments. We replaced FDPs with FPDs.

Table 3: the values present are negative. In a first sight, the reviewer thought that the negative value meant that the loss was negative, which means, a bone gain. After some thought, I realized that the value meant the actual loss. I suggest a fine tuning in the table to avoid misinterpretation (e.g., add information in the table description).

As you mentioned, we changed the negative to positive to avoid confusion. Thank you.

The survival rate obtained in the present study is superior to the average survival rate described in the literature. Seeing the exclusion criteria (especially bruxism), this finding makes sense. No subgroups inside IT and IU, regarding MBL, were shown. The authors claimed that no single factor was found to have a significant effect on MBL. Nevertheless, it would be useful (as supplementary material) to have the values of subgroup division.

Thanks for the comments. We added the results of linear mixed model analysis as supplementary table.

Figure 2: Add A and B to the figures to follow the legend. In addition, I suggest pointing out (with an arrow) the fracture or adding a zoomed picture to facilitate the appreciation of the fracture.

I inserted arrows to indicate the fractured abutment screw and implant. Thanks your for your advice.

Discussion

Authors discussed the causes of implant failure, lack of osseointegration, fractures, peri-implantitis and non-functional implants. The discussion is well-written and tackled the reason and literature comparison. However, In the reviewer’s view, the discussion is significantly more focused on implant failures than MBL. Since the objective of the present study embraces both outcomes, I suggest the addition of a paragraph about MBL.

We added a little part about marginal bone loss in the discussion section. Due to the difference of recall length, it was not easy to compare the amount of MBL between the two implant types. We’re sorry about that.

Implant fractures. I suggest disclaiming if patients have some non-diagnosed parafunctional habit (e.g., just clenching, without grinding) and, after the first fracture, if some action was taken (e.g., bite guard or Botox application).

Thanks for the comments. We are not offering patients night guard or botox injection when the patient had strong mastication muscles. We’ll consider bite (night) guard afterwards.

Reviewer 2 Report

Introduction:

- The state of the art on the article topic si described, but a general description on why the implant next is important is missing. For this propose I suggest you to discuss and cite a recent published article that analyzed the point in details. 

Carossa, M.; Alovisi, M.; Crupi, A.; Ambrogio, G.; Pera, F. Full-Arch Rehabilitation Using Trans-Mucosal Tissue-Level Implants with and without Implant-Abutment Units: A Case Report. Dent. J. 202210, 116. https://doi.org/10.3390/dj10070116

- Add the study hypotheses after the aim of the study

Materials and methods:

- the exclusion criteria are currently well described, but the inclusion criteria can be better explained. Please describe the inclusion criteria in a more accurate way to guarantee the reproducibility of the study (example single or multi unit implants supported prostheses, implants with platform switching and implants with...). Please be as more accurate as possible. 

- in the exclusion criteria you should state that you excluded all the implants design that differs from the one you considered. 

- line 114: ''Implant survival was defined as follows: the implant remained in the patient's mouth, and  the restoration functioned normally during the last periodic visit. Therefore, removed and  buried (submerged and not functional) implants were classified as failures.'' Please add references supporting these definitions that you adopted. 

- Please described how you evaluated periimplantitis

- Information about the crown/bridge materials are missing. Please add them in the M&M section if they were decided prior to the study or in the results section if you called data retrospectively. 

- We discover only in the results section that same implants had graft procedures that led to failures. Please add the graft information in the M&M section or in the results section (this makes you also understand the importance to be as more accurate as possible in the inclusion/exclusion criteria).

- also add information about the load of the implants. Were they all delayed loaded? Did you follow a one stage or two stage techniques? You have to describe all this informations. 

- there are very few informations about the soft tissue. Did you evaluate peri-implant probing depth, bleeding on probing and plaque index? these are all factors that may be related to changes in the neck of the implant (the part that is probably in contact with the soft tissue)

Results: add the incidence of periimplantitis and any information about the above mentioned soft tissue parameters

Discussion:

- Discuss if the study hypotheses were accepted or rejected accordingly to the results

- you currently discussed the implant survival rate comparing it with other studies. But a comparison of the marginal bone loss with other implant designs is missed. For this propose, please discuss and cite this paper doi: 10.1111/cid.13113 . It is important to discuss also the MBL and not only the survival rate. 

- What is the rational to perform a study comparing 6 years follow up with a 15 years follow up? do you think that it would be better to compared to the two variables at the same follow up (6 years for the present study)? please discuss this point in the discussion.

Author Response

Dear reviewer. First of all, thanks for the review. I really appreciated. Some rephrased part may not meet your standard due to the nature of retrospective study. Soft tissue information was not fully recorded. The recall extensions were different between the IU and IT implant. However, we tried to follow your comments and advices as possible.

Introduction

- The state of the art on the article topic is described, but a general description on why the implant next is important is missing. For this propose I suggest you to discuss and cite a recent published article that analyzed the point in details. 

Carossa, M.; Alovisi, M.; Crupi, A.; Ambrogio, G.; Pera, F. Full-Arch Rehabilitation Using Trans-Mucosal Tissue-Level Implants with and without Implant-Abutment Units: A Case Report. Dent. J. 202210, 116. https://doi.org/10.3390/dj1007011

I cited the reference you recommended. Thanks

Add the study hypotheses after the aim of the study

As you advised, I added study hypothesis followed by the aim of the current study

Materials and methods

the exclusion criteria are currently well described, but the inclusion criteria can be better explained. Please describe the inclusion criteria in a more accurate way to guarantee the reproducibility of the study (example single or multi unit implants supported prostheses, implants with platform switching and implants with...). Please be as more accurate as possible. 

Inclusion criteria were that: (1) all patients met the diagnostic criteria for a dentition defect; (2) the patients had no contraindications to surgery; (3) informed consent was provided; and (4) female participants were non-pregnant, non-lactating, and not menstruating. 

I added inclusion criteria as you advised.

  • in the exclusion criteria you should state that you excluded all the implants design that differs from the one you considered.

I also added the exclusion criteria you mentioned

  • line 114: ''Implant survival was defined as follows: the implant remained in the patient's mouth, andthe restoration functioned normally during the last periodic visit. Therefore, removed and  buried (submerged and not functional) implants were classified as failures.'' Please add references supporting these definitions that you adopted. 

I could not find appropriate study which classified buried (not used) implants as failures. I included the buried implant (one implant) as failure because the implant could not continue it’s function. Maybe, more experienced clinician could remove the fractured part and continue to use the implant. In such situation, the implant cannot be classified as a failure. I also have almost 28 years of implant practice experience and I used to remove the abutment or screw fractured parts. However, the case was not possible for me to remove the part, therefore, I decided to bury the implant. Most of all, I re-fabricated a restoration (FDP) using an additional implant, so I classified the buried implant as failure

  • Please described how you evaluated periimplantitis

Thanks for the comments. I added a paragraph about “the diagnosis of peri-implantitis oil the materials and methods section. Unfortunately, we did not have all the probing records, therefore, we used the bone level data and BOP data to clinically diagnose peri-implantitis.

  • Information about the crown/bridge materials are missing. Please add them in the M&M section if they were decided prior to the study or in the results section if you called data retrospectively.

Unfortunately, we do not have all the data about restorative materials.

  • We discover only in the results section that same implants had graft procedures that led to failures. Please add the graft information in the M&M section or in the results section (this makes you also understand the importance to be as more accurate as possible in the inclusion/exclusion criteria).

Thanks for your comments. I added informations about the graft material but I did not present the commercial name, which was BioOss. Please let me know if the commercial name is necessary.

  • also add information about the load of the implants. Were they all delayed loaded? Did you follow a one stage or two stage techniques? You have to describe all this informations.

Thanks for the comments. We included all one staged (non-submerged) and two staged (submerged and received recovery surgery afterwards) approached. We thought 1-stage or 2-stage does not affect the amount of marginal bone loss, therefore, we did not address the way of surgery in the materials and methods section. However, we included only delayed loaded implant in the current study, because immediate loading may not produced increased amount of marginal bone loss, but immediate loading may increase the failure rate. We added this in the exclusion criteria. Thanks again for your advice.

- there are very few informations about the soft tissue. Did you evaluate peri-implant probing depth, bleeding on probing and plaque index? these are all factors that may be related to changes in the neck of the implant (the part that is probably in contact with the soft tissue)

Thanks for the comment. Unfortunately, we did not record soft tissue information for all the implants. We just recorded bleeding on probing for implant with increased marginal bone loss, which were usually 2mm compared to the initial bone level. These are the weakness of our study. Now we are keeping those information for future studies. I’m sorry about that.

Results

: add the incidence of periimplantitis and any information about the above mentioned soft tissue parameters

Unfortunately, we did not detailed soft tissue informations for all the implants. This is the weakness of the current study. Because the plaque index and probing depth were not recorded for all implants, peri-implantitis can be diagnosed using the case of more than 3 mm of marginal bone loss compared to the initial bone level combined with bleeding on probing. Based on the above, the incidence of peri-implantitis was 2.0% at 4 years, and 2.8% at 6+ years recall. These results were very different to the previous studies. We assumed that

Discussion

- Discuss if the study hypotheses were accepted or rejected accordingly to the results

We added the result of hypothesis in the beginning of the discussion.

  • you currently discussed the implant survival rate comparing it with other studies. But a comparison of the marginal bone loss with other implant designs is missed. For this propose, please discuss and cite this paperdoi: 10.1111/cid.13113 . It is important to discuss also the MBL and not only the survival rate. 

Thank you for the valuable information about the reference. We added the reference in the introduction part, and added a short sentence about the MBL in the discussion section

  • What is the rational to perform a study comparing 6 years follow up with a 15 years follow up? do you think that it would be better to compared to the two variables at the same follow up (6 years for the present study)? please discuss this point in the discussion.

Yes, you are right. The IU implant was lately introduced in the market, we could not have long term clinical recall data. Although a direct comparison between IT and IU implants was not possible, we could report long-term clinical results for IT implants. We thought that was also meaningful.  We mentioned the problem as a limitation of this study in the last section of discussion.

Round 2

Reviewer 2 Report

Dear Authors,

thank you for answering and to clarify all my comments.